# Aggregation-Induced Emission Luminogens for Enhanced Photodynamic Therapy: From Organelle Targeting to Tumor Targeting

**DOI:** 10.3390/bios12111027

**Published:** 2022-11-16

**Authors:** Jiahe Zhou, Fen Qi, Yuncong Chen, Shuren Zhang, Xiaoxue Zheng, Weijiang He, Zijian Guo

**Affiliations:** 1State Key Laboratory of Coordination Chemistry, School of Chemistry and Chemical Engineering, Nanjing University, Nanjing 210023, China; 2Chemistry and Biomedicine Innovation Center (ChemBIC), Nanjing University, Nanjing 210023, China; 3Nanchuang (Jiangsu) Institute of Chemistry and Health, Nanjing 210000, China

**Keywords:** organelle targeting, photodynamic therapy, aggregation-induced emission, tumor targeting

## Abstract

Photodynamic therapy (PDT) has attracted much attention in the field of anticancer treatment. However, PDT has to face challenges, such as aggregation caused by quenching of reactive oxygen species (ROS), and short ^1^O_2_ lifetime, which lead to unsatisfactory therapeutic effect. Aggregation-induced emission luminogen (AIEgens)-based photosensitizers (PSs) showed enhanced ROS generation upon aggregation, which showed great potential for hypoxic tumor treatment with enhanced PDT effect. In this review, we summarized the design strategies and applications of AIEgen-based PSs with improved PDT efficacy since 2019. Firstly, we introduce the research background and some basic knowledge in the related field. Secondly, the recent approaches of AIEgen-based PSs for enhanced PDT are summarized in two categories: (1) organelle-targeting PSs that could cause direct damage to organelles to enhance PDT effects, and (2) PSs with tumor-targeting abilities to selectively suppress tumor growth and reduce side effects. Finally, current challenges and future opportunities are discussed. We hope this review can offer new insights and inspirations for the development of AIEgen-based PSs for better PDT effect.

## 1. Introduction

Cancer, as one of the most fatal diseases, caused nearly 10 million deaths in 2020, as reported by the World Health Organization International Agency for Research on Cancer (IARC) [1]. According to the Chinese National Cancer Center, the survival rate has increased by 10% to 40.5% compared to 10 years ago. This surprising result could be attributed to the emergence of various therapies to overcome cancer [2]. At present, cancer treatment methods mainly include surgery, radiotherapy, chemotherapy, immunotherapy, etc. [3,4,5]. However, these therapies still come with limitations, such as nausea and vomiting, which are common side effects of chemotherapy and radiation [6], and the effectiveness of immunotherapy for specific tumors [7]. In recent years, people have gradually turned their attention to the field of phototherapy [4,8,9,10,11,12,13,14], which shows various attractive features such as non-invasiveness, low toxicity and good biocompatibility [11,15]. 

From the perspective of photodynamic therapy (PDT) [16], photosensitizers (PSs) produce ROS to kill tumor cells under light irradiation. PSs generate ROS through two processes [8,11,17,18,19,20]: (a) Type I: through proton or electron transfer [21], direct reaction with substrate or solvent molecules to form O_2_^•−^ and •OH [22]; (b) Type II: the triplet PSs transfer their energy to triplet oxygen molecules to form ^1^O_2_ [10]. Nevertheless, PDT suffers from problems such as aggregation in aqueous solutions, which leads to both fluorescence quenching (ACQ effect) [23,24] and ROS quenching. In addition, the short lifetime of ^1^O_2_ shortens its effective radius, which might limit the Type II PDT efficacy [25]. The PSs with Type I ROS production capacity will be described in this paper, and the remaining examples without additional description are those of Type II ROS or ROS species not described in the original literature. Aggregation-induced emission (AIE) was proposed to inhibit PSs aggregation and promote ROS production. On the other hand, organelle-targeting strategies could generate ROS in situ and cause severe damage to organelles, which could help to solve the problem of short ^1^O_2_ lifetime [26,27,28]. In addition, tumor-targeting strategies can effectively improve the specificity of PDT and reduce side effects, which can help improve PDT performance [28,29,30,31].

In this review, we summarized current approaches to enhance photodynamic therapy based on AIE from two different perspectives (Figure 1). First, the design strategies and antitumor applications of AIEgen-based PSs that are able to target cell plasma membranes, mitochondria, lysosomes, lipid droplets, nuclei, the endoplasmic reticulum, and the Golgi apparatus were introduced. Second, different approaches for tumor-targeting AIEgen-based PSs were covered. This review does not aim to be comprehensive; readers are also encouraged to refer to other excellent reviews for more information [32,33,34,35,36,37]. Due to the fast advances in this field, we only summarized some representative works during the past three years. We hope that this review can provide some ideas for related research and promote the progress of PDT therapy.

## 2. Organelle-Targeting AIE-PDT

PSs should be designed to be intracellularly targetable due to high ROS reactivity and short diffusion distance, which may enhance PDT efficacy significantly. Currently, many AIEgen-based PSs that showed subcellular organelle-targeting abilities have been reported. Most of them were constructed to target mitochondria, lysosomes, lipid droplets, and plasma membranes because the targeting strategies were well-established [28]. AIEgen-based PSs targeting other organelles, such as the endoplasmic reticulum (ER) and Golgi apparatus (GA), were less reported. 

### 2.1. Membrane Targeting

In addition to secreting and transporting proteins, the cell membrane also absorbs and excretes substances inside and outside through endocytosis and exocytosis. If the cell membrane is damaged, this results in the increased permeability of tumor cells. It may promote the uptake of deleterious material [38,39,40,41]. There are two main strategies for targeting cell membranes [28]. One is to build a positive charge structure based on a similar method of targeting mitochondria, and the introduction of amphipathic structures is able to obtain stronger cell-membrane-targeting capabilities [42]. The other strategy is using membrane-specific ligand modification [43]. The antigen-modified PS, for example, can specifically recognize receptors abundantly expressed on tumor membranes. 

Recently, Tang et al. designed and synthesized AIE-PSs with NIR-emission (735 nm) that could target cell membranes [38]. They infused the cationation structure into TBMPEI, triggering the cell-membrane-targeting characterization (Figure 2). The fluorescence signals of DCFH incubated with TBMPEI significantly increased nearly 900 times in 80 s under white light, indicating remarkable ROS generation ability. The species of ROS produced by TBMPEI were shown to be a mixture of Type I and Type II. After being stained with TBMPEI, multiple cells showed high Pearson’s correlation coefficients and high signal-to-noise ratio. They also confirmed that TBMPEI had good cytotoxicity through the IC50 values of 4T1, A549 and Hela cells under light excitation. The researchers found that the integrity of the cell membrane was destroyed, and DNA degradation even occurred. TBMPEI was shown to induce necroptosis of tumor cells by targeting cell membranes with the Annexin V-FITC test and tumor section. The tumor in vivo had been inhibited significantly in the experiment group treated with TBMPEI.

Recently, Liu et al. developed a cancer immunotherapy that causes pyroptosis through a membrane-targeted photosensitizer TBD-3C with AIE characteristics (Figure 3) [40]. In addition to stimulating macrophage M1 polarization, it can cause maturation of dendritic cells (DCs) and activate CD8 + cytotoxic T lymphocytes (CTLs) (Figure 3). The result of flow cytometry showed that TBD-3C is capable of successful membrane anchoring. Flow cytometry and DCFH-DA upon light irradiation (40 mW cm^−2^ for 10 min) were further used to confirm the ROS generation ability of TBD-3C in pancreatic cancer. After irradiation, TBD-3C induced swelling on KPC and panc02 cells, showing the typical morphological features of pyroptosis. KPC and Panc02 cells treated with TBD-3C release lactate dehydrogenase (LDH) as the typical signal of pyroptosis. This study developed a pyroptosis-based photodynamic anti-tumor immunotherapy approach.

Yang et al. reported an amphiphilic perylene derivative AIE-PSs named AP (*ϕ***_Δ_** = 0.22, methylene blue as reference) which is capable of cell-membrane-targeting [44]. AP could form nanoparticles by self-assembly in an aqueous solution and decompose into free monomeric molecules after membrane anchoring (Figure 4). CLSM images of MCF-7 cells stained with AP were clearly eliminated. H&E staining and TUNEL images indicates that apoptosis occurred in many tumor cells. In addition, in vivo biocompatibility tests revealed that AP was almost nontoxic.

### 2.2. Lysosomal Targeting

As the main digestive site in cells, lysosomes degrade intruding toxic substances using more than 60 hydrolytic enzymes, making them crucial in autophagy and secretion [28,34,45]. Lysosome-targeted strategies have also become instructive and meaningful for PDT therapy. There are two main approaches: (1) Most examples target lysosomes through the modification of lipophilic amines with the addition of morpholines and other amine groups [46,47,48,49,50,51,52,53,54,55,56,57,58,59,60]. (2) A promotion of endocytosis can also transport PSs from the endosome into the cell and capture them in the lysosome [61,62,63,64,65,66,67,68,69,70].

According to Niu et al., a BTZPP molecule with A-D-A structure was synthesized and the PDT effect was investigated (Figure 5) [47]. It had been demonstrated in experiments that BTZPP NPs are capable of high ^1^O_2_ quantum yield (72.3%, rose bengal as a reference), NIR emission (635 nm) and good photostability under harsh conditions such as acidity. Moreover, BTZPP with LysoTracker Green had a good Pearson coefficient (0.91). Apoptosis was observed in Hela cells treated with BTZPP under light conditions. Based on MTT assay, BTZPP NPs showed IC_50_ value. In addition, BTZPP NPs exhibit the characteristic of long-time in vivo imaging, which is helpful for diagnosis. Meanwhile, the in vivo experiment indicated BTZPP had no obvious systemic toxicity.

Tang et al. synthesized a series of molecules with varying amounts of thiophene spacers [70]. As the first single molecules to deliver all phototheranostics, TSSIs can effectively deliver fluorescence and photoacoustic imaging, such as modalities including PDT, photothermal imaging (PTI), photothermal therapy (PTT) and so on (Figure 6). The TSSI NPs simultaneously exhibit NIR-II emission (1000 nm), high ROS generation (Type I), and good photothermal conversion efficiency (46%). TSSIs have excellent ROS generation ability, with the emission intensity of DCFH increased over 250-fold after irradiation. The generation of Type I ROS made the PDT effect more effective under hypoxic conditions. TSSI showed good lysosomal targeting ability, and the Pearson coefficient was up to 0.964. Additionally, the authors demonstrated that TSSI NPs are ingested by cells using energy-dependent endocytosis. H&E staining of tumor slices revealed that tumor cell apoptosis occurred under NIR irradiation. Moreover, compared with the control group, tumor growth was significantly inhibited until extinct with TSSI treatment. Meanwhile, no significant systemic toxicity was discovered.

Similarly, the authors reported another example, TTT-4, with better photoacoustic-guided imaging ability in 2021 (Figure 7) [59] and a better therapeutic effect on tumor tissue with a powerful lysosomal targeting ability. TTT-4 also generates Type I ROS with high generation ability. The fluorescence of DCFH increased 160-fold under white light (22.1 mW/cm^2^). MTT assay indicated that the IC50 value of 4T1 cells incubated with TTT-4 was less than 1 μmol, which should be attributed to the excellent effects of PDT and PTT. Furthermore, H&E staining of tumor slices revealed the tumor tissue exhibited lots of apoptotic cells. 

### 2.3. Mitochondrion Targeting

The mitochondria play vital roles in energy production and intracellular signal transmission in cells [71]. The overexpression of ROS in mitochondria may cause oxidative stress, which may disrupt the mitochondrial microenvironment and lead to apoptosis, autophagy and necroptosis [72,73,74]. Consequently, the mitochondrion is a suitable organelle for PDT. In order to realize mitochondrial localization, several strategies have been explored, including: (1) introducing lipophilic cations structures that are intrinsically or modified [75,76,77,78,79,80,81,82,83,84,85,86,87,88,89,90,91,92,93,94,95,96,97,98,99,100,101,102,103,104,105,106,107,108,109]; and (2) using mitochondria-specific peptides [110,111,112,113]. 

Zheng et al. developed a series of positively charged AIEgens (Figure 8) [71]. DCQu is capable of specific mitochondrial targeting with high ^1^O_2_ generation efficiency and NIR emission. DCPy and DCQu had superior AIE properties compared to other compounds. Following co-incubation of DCQu and H2DCF-DA with Hela cells, fluorescence imaging experiments revealed that DCQu generates ^1^O_2_ efficiently (2.1-fold higher than DCPy) during irradiation. DCQu exhibits high Pearson’s correlation coefficients of 0.95, indicating superior specificity for mitochondrial staining. Furthermore, MTT assay showed that DCQu had a good therapeutic effect on tumor tissues. Using hematoxylin and eosin (H&E) analysis, it was evident that many cells with highly condensed nuclei were apoptotic. Compared with the control group, the survival rate of mice treated with DCQu increased significantly.

Tang et al. drew on a cationization strategy to synthesize DTPAN, DTPAPy, DTPANPF_6_ and DTPAPyPF_6_ (Figure 9) [78]. Using injected cationation with a strong ICT effect, DTPANPF_6_ and DTPAPyPF_6_ are endowed with high Type I radical production capacity and the ability to target mitochondria. The HPF intensity enhancement of DTPANPF_6_ and DTPAPyPF_6_ was 37.4- and 30.0-fold under irradiation (20 mW cm^−2^). Both DTPANPF_6_ and DTPAPyPF_6_ exhibited high Pearson coefficients for both Hela and MCF-7 cells. The viability of HeLa cells suggests they both have therapeutic potential under hypoxic conditions. H&E staining suggested prominent cell necroptosis occurred in tumor tissues. In in vivo PDT experiments, tumor growth was significantly inhibited when treated with DTPANPF_6_ and DTPAPyPF_6_. 

Tang et al. reported another study on PDT treatment of mitochondrial targeting to initiate autophagy (Figure 10) [90]. TACQ exhibits near-infrared emission (635 nm), high photothermal conversion efficiency (55%), and high ^1^O_2_ generation capacity. The reduction of absorbance for ABDA treated with TACP at 378 nm after irradiation for 60 s reached 90.3%. The quinoline cation of TACQ selectively accumulated in the mitochondria. The Pearson coefficient of TACQ with Mito-Tracker Green (MTG) was as high as 0.95. In comparison with MTG, TACQ provided better 3D mitochondrial images with higher lateral resolution. TEM characterization of HeLa cells indicates that autophagosomes are formed inducing mitophagy after TACQ treatment. The authors suggest that TACQ accelerates mitochondrial instability and leads to apoptosis in cancer cells.

### 2.4. Lipid Droplet Targeting

Lipid droplets are lipid-rich organelles found mainly in adipose tissue. They are highly dynamic organelles involved in intracellular lipid storage, metabolism, and membrane transfer. Additionally, LDs are being pursued as a target of PDT [114,115,116,117,118,119,120]. The LD-targeted photosensitizer should have a highly hydrophobic structure and be less polar than the other fractions in the cell [121,122,123].

Dai et al. designed and synthesized an AIE-PSS (TTI) with strong lipophilic and near-infrared emission (NIR) (Figure 11) [119]. The ^1^O_2_ quantum yield of TTI, was determined to be 85.16% using an equation. In addition, the calculated Clog *p* values of TTI and its derivatives ranged from 8.3 to 9.4, all within the range of 4.5 to 9.5 which could target LDs. The Pearson coefficient of TTI with BODIPY 493/503 was calculated to be 0.9491. Cell apoptosis was detected by Annexin V-FITC/PI co-staining. The results showed that TTI could effectively induce apoptosis of HepG2 cells under white light irradiation. 

Tang et al. synthesized two AIE-PSs (PI and PTI) with near-infrared (NIR) emission properties and the ability to specifically target lipid droplets (Figure 12) [124]. PTI was obtained by introducing a thiophene ring into the PI skeleton to enhance the ISC process. The authors confirmed that targeting lipid droplets using PI and PTI caused ferroptosis by monitoring intracellular glutathione (GSH) and glutathione peroxidase 4 (GPX4) levels. Furthermore, it is important to note that the authors used homologous MCF-7 cell membranes to wrap the PLGA core, composed of PTI and PLGA, to achieve a homologous targeting ability. Besides, PTI has a distinguished ROS generation ability with the intensity of H2DCFH-DA in PBS, increasing 120-fold upon white light irradiation (50 mW cm^−2^). The synthesized MCFCNPs have a good inhibitory effect on tumors in vivo without obvious toxic side effects.

Similarly, Tang et al. reported another case in 2021 (Figure 13) [114]. They combined the targeted LDs AIE-PSs (MeTIND-4) with DC cell membranes to achieve antigen functioning as a biomimetic nano-photosensitizer (DC@AIEdots). While the exogenous cell membrane stimulates the proliferation and activation of T cells, internal AIE photosensitizers target tumor cells for PDT. The fluorescence intensity of DCFH treated with MeTIND-4 increased striking by 600-fold after irradiation (60 mW cm^−2^). Therefore, PSs not only produces sufficient ROS to eliminate tumor cells, but also promotes immunogenic cell death. In addition, the efficiency of the tumor delivery of photosensitizers had been effectively improved (1.6 times). DC@AIEdots can not only kill in situ tumors, but also suppress distant tumors by activating the immune system against tumor growth. This work provides a significant guide for the development of related fields.

In another example reported by Liu et al., they constructed NIR-emitting PSs (TPET-IS, TPET-FU and TPEF-IS) with the function of targeting LDs [125]. Based on the theoretical calculations, the Log *p* values for TPET-IS, TPET-FU and TPEF-IS were 9.39, 7.89 and 8.03, respectively, which were higher than BODIPY 493/503 (a commercial LD marker), indicating a good LD-specific targeting. The Pearson’s correlation coefficients were 0.94, 0.96, and 0.97. It is worth noting that the survival rate of Hela cells at 50 μm concentration of the three compounds was more than 90% under dark conditions.

### 2.5. Endoplasmic Reticulum Targeting

ROS-induced stress in the endoplasmic reticulum (ER) may lead to the activation of downstream immune pathways, resulting in immunogenic death of cells [28]. PSs modified by specific peptides or methyl sulfonamide usually have ER-targeting abilities, and some ring metal complexes can also target the ER [126]. 

Based on the reported AIE material TBP, Su et al. grew, in sulfonic acid, functional groups through a cation strategy to prepare TBP-SO_3_ to obtain the ability to target the ER (Figure 14) [127]. TBP-SO_3_ exhibited high Type I ROS generation capability, while the fluorescence spectra of DHR123 treated with TBP-SO_3_ increased nearly 800-fold after irradiation (23.4 mW/cm^2^). In CLSM co-localization assay, the Pearson coefficient of HeLa cells incubated with ER-Tracker Red (targeting ER) and TBP-SO_3_ was 0.93. It was found that the cell survival rate was not significantly decreased at 30 μmol concentration without light. The IC_50_ value of TBP-SO_3_ under white light irradiation was less than 5 μmol. The above experiments showed that TBP-SO_3_ had a good application prospect in PDT.

Additionally, Tang et al. reported two ER-targeting Type I AIE-PSs (*α*-TPA-PIO and *β*-TPA-PIO) in 2020 (Figure 15) [128]. The results showed that *β*-TPA-PIO inhibited tumor cell growth under hypoxic conditions. Images from colocalization experiments showed good overlap between *β*-TPA-PIO and ER. The fluorescence signal of HPF containing *α*-TPA-PIO or *β*-TPA-PIO increased 6- and 11-fold after white light irradiation (20 mW cm^−2^). An in vitro study including co-localization, Western blot, and immunohistology analyses found that PSC could lead to autophagy and apoptosis by inducing ER stress. Additionally, in vivo experiments indicated that *β*-TPA-PIO was effective in eliminate solid tumors. Researchers suggest that PIO induces immunogenic cell death, facilitating the combined effects of PDT and immunotherapy.

### 2.6. Golgi Apparatus Targeting

It has been reported for the first time that photosensitizers with AIE characterizes can target the Golgi apparatus (GA), as shown by Guo et al. (Figure 16) [129]. They synthesized and found that TPE-PYT-CPS has ER-targeting capability via caveolin/raft endocytosis. By utilizing structure–activity relationships, researchers believe cyano-induced rod-like packing in molecules plays a key role in GA targeting. Pyrene units and cyano-pyridinium salt moiety have been shown to reduce the energy gap (ΔE_ST_) between the lowest singlet state (S_1_) and the lowest triplet state (T_1_), so as to promote the generation of ROS. The release of ROS causes oxidative stress and damages the GA. Then, the structural protein p115 is cleaved into N-terminal and C-terminal fragments, which are then transported into the nucleus and up-regulate apoptosis proteins p53, triggering mitochondrial dysfunction and leading to apoptosis. The decomposition rate (*k*_d_) of ABDA in an aqueous solution of TPE-PYT-CPS, which represents ROS generation ability, was 32.85 nmol per minute. Specifically, the intracellular ^1^O_2_ was detected using the CLSM method, and the image showed obvious green fluorescence. A Pearson correlation coefficient of 0.98 indicated that TPE-Pyt-CPS had an excellent GA targeting ability. After incubation with TPE-PYT-CPS (0.2 μm), flow cytometry showed 56.7% apoptosis in HeLa cells. The present study provides a ground-breaking report on a promising AIE-enhanced PDT strategy, whose design has important implications for related GA-targeted photosensitizers.

### 2.7. Nucleus Targeting

As the “brain” in the cell, the nucleus is responsible for DNA storage, regulating cell metabolism, intracellular signaling and regulating the cell cycle [28]. The nuclear pore is located in the nuclear envelope and is approximately 40 nm in diameter, allowing some water-soluble small molecules to freely traverse the nucleus. Macromolecules such as proteins and RNA require energy and transporters to enter the nucleus. Generally, PSs target the nucleus in two ways: (1) modification of short peptide chains with nuclear targeting capability [60,61,130,131]; and (2) aptamer modification [132,133,134,135].

Recently, Mao et al. developed an AIE-PSs (MeTPAE) with nuclear targeting capability based on a triphenylamine framework (Figure 17) [130]. MeTPAE can not only combine with histone deacetylases (HDACs) to inhibit cell proliferation, but also be synergistically treated with PDT. Additionally, MeTPAE not only has high ROS generation, including Type I and Type II ROS (*Φ*_Δ_ = 77.2% in water), but its excellent two-photon absorption property also provides convenience for PDT. The fluorescence intensity of MeTPAE is further enhanced after binding to nucleic acid through electrostatic interaction and hydrogen bonding. Additionally, the Pearson coefficient of MeTPAE versus Hoechst 33342 was 0.85. Moreover, MeTPAE binding to telomeric G4 DNA caused efficient destruction of nucleic acids and inhibited telomerase activity in nucleic acid titration experiments.

Tang et al, developed the first AIE-PSs (TPE-4EP+) that can monitor its own photodynamic therapy response in real time in situ (Figure 18). It has an extremely high singlet oxygen production efficiency (ABDA decomposition rate of TPE-4EP+ reached 118.5 nmol min^−1^ under 4.2 mW/cm^2^ white light) and undergoes a process of transfer from mitochondria to the nucleus during the induction of apoptosis. The authors believe that this is because the charged TPE-4EP+ gradually dissociates from the binding to the mitochondrial membrane due to the loss of mitochondrial membrane potential during apoptosis. Moreover, due to the expansion of nuclear membrane permeability, it binds to a large number of DNA in the nucleus through electrostatic adsorption and illuminates the nucleus.

### 2.8. Multiple Organelle Targeting

In terms of multi-organelle targeting, a single AIE-PS has multiple-organelle-targeting capabilities by structural design. Another strategy is the use of different AIE-PSs targeting various organelles to generate ROS and damage organelles. At the same drug concentration, the therapeutic effect of drugs with the ability to target multiple organelles is better than that of drugs only target one specific organelle. This can effectively improve the therapeutic effect of PDT and reduce the use of drug concentration and the toxic side effects. Therefore, the development of PSs with multiple targeting sites has attracted much attention [41,136,137,138,139,140,141].

Tang et al. encapsulated TTFMN by introducing triphenylethylene units on the basis of TFMN [61]. To facilitate the PDT antitumor effect of TTFMN, a pH-activated TAT peptide-modified amphiphilic polymer encapsulates TTFMN to transport PSs into the tumor nucleus (Figure 19). Both TFMN and TTFMN emission wavelengths reached NIR (651 nm). Moreover, these two compounds had great application potential as they all produced ROS through Type I mechanisms (intensity of DCFH containing TTFMN was enhanced by nearly 500-fold under 22.1 mW cm^−2^ white light irradiation in 5 min). After entering the cell, TTFMN first enters the lysosome, where it is activated by acid and then transported to the nucleus. With the growth of incubation time, some TTFMN-NPs translocated to the perinuclear region. When the time reached 12 h, a large number of TTFMN-NPs crossed the perinuclear region and even partially entered the nucleus. The apoptosis of tumor cells induced by TTFMN-NPs was confirmed by terminal deoxynucleotidyl transferase-mediated nick end labeling (TUNEL) staining. In all organs of mice treated with TTFMN-NP, H&E-stained slices showed no significant organ damage. 

After that, the authors reported another modified AIE-PS for nuclear targeting (Figure 20) [60]. TPE-TTMN-TPA was synthesized by infusing a diphenylamine structure on the basis of TTMN. Compared with the previous work, the TPE-TTMN-TPA nanoparticles had more red-shifted NIR emissions, higher Type I ROS generation capacity (intensity of DCFH enhanced nearly 600-fold after 10 min of 22.1 mW cm^−2^ white light irradiation) and better nuclear-targeted delivery. T4-NPs composed of TPE-TTMN-TPA and SA-TAT also needed to be activated by lysosomal acid. After incubation for 1 h, the Pearson coefficient of T4-NPs and lysosomes reached 0.94, while it decreased to 0.62 after 6 h, proving its effective escape from lysosomes. As shown in the image, more and more T4-NPs entered the nucleus as time elapsed. Flow cytometry analysis showed that a large number of tumor cells underwent apoptosis after photoexcitation, which proved the effectiveness of the PDT effect.

Another example is the single AIE-PS (TBZPy, MTBZPy, TNZPy, MTNZPy) with lysosomal and mitochondrial targeting capabilities reported by Tang et al., (Figure 21) [138]. By constructing strong intramolecular charge transfer (ICT), the electron-rich system can facilitate the progress of Type I PDT by providing electrons. In the presence of TNZPy, they used H2DCF-DA as an indicator to define the ROS generation where the intensity increased 140-fold after white light irradiation (50 mW cm^−2^). The Pearson coefficients of TNZPy for lysosomes and mitochondria were 0.81 and 0.88, respectively. With the development of time, the Pearson coefficient of lysosomes gradually decreased, and the corresponding mitochondrial gradually increased. This suggests that TNZPy can efficiently escape from lysosomes and accumulate in mitochondria. The authors suggest that ROS generation after photoexcitation synergistically destroys lysosomes and mitochondrial organelles to induce apoptosis. The IC_50_ value of TNZPy for Hela cells was less than 6μmol under both hypoxic and normoxic conditions, indicating potential for treating tumors under hypoxia conditions. In the in vivo experiment, the body weight of tumor-bearing mice did not change significantly, while the tumor growth was significantly inhibited.

In 2020, Tang et al. proposed a pioneering strategy of 1 + 1 + 1 > 3 (Figure 22) [41]. They synthesized a series of different AIE-PSs (TFPy, TFVP and TPE-TFPy) but with the same skeletal structure through subtle structural adjustments, and they were able to specifically anchor to mitochondria, cell membranes, and lysosomes to damage organelles by producing ROS. The ^1^O_2_ quantum yield of TFPy, TFVP and TPE-TFPy compared with rose bengal were 25.2%, 18.3% and 63.0%, respectively, suggesting good ROS production. In situ, TPE-TFPy aggregates easily form nanosized aggregates that endocytose into lysosomes and specifically illuminate them. The positively charged pyridine moiety of TFPy may bind to mitochondria, leading to mitochondrial targeting capability. In part, TFVP’s lower membrane permeability coefficient may be due to its higher free-energy barrier, which confers its specific aggregation properties on the cell membrane. Fluorescence imaging results showed that TFPy, TFVP and TPE-TFPY showed strong targeting ability towards mitochondria, cell membranes and lysosomes, respectively. Compared with the single photosensitizer treatment, the combination of the three treatments significantly enhanced the anti-tumor effect of PDT. Notably, the synergistic treatment of the three did not affect their biocompatibility.

## 3. Tumor-Targeting AIE-PDT

With the continuous research of anti-tumor drugs and the rapid development of tumor biology, PSs are usually encapsulated into nanoparticles to enhance their absorption in the biological environment and obtain better targeting ability [142,143]. For nanoparticles (NPs), tumor-targeting strategies are divided into active-targeting and passive-targeting strategies [144,145,146]. Nanomedicine can extravasate and remained in the pathological site mainly based on the enhanced permeability and retention (EPR) effect, which can be classified as passive targeting [147,148]. However, the EPR effect can only deliver very limited amounts of PS to tumor tissues and its efficiency has been challenged in recent years. In this regard, some active targeting strategies have been proposed to help enhance tumor-targeting ability, such as the modification of some ligands that specifically bind to tumor-overexpressed receptors, and encapsulating PSs into some engineered cell membranes as camouflage for tumor targeting, etc.

### 3.1. Passive Targeting

Generally, passive targeting is based on the EPR effect, optimizing the size or surface properties of nanoparticles [143,149]. Nano-systems (20–200 nm) can selectively penetrate tumor stroma via newly formed leaky vessels [150,151,152]. Compared with free drug molecules, NPs are preferentially accumulated at tumor sites through the EPR effect. There are two main ways to enhance the EPR effect: (1) A greater penetration of NPs through the extracellular matrix (ECM) could improve the EPR effect [144,153,154]. Injecting hyaluronidase (HAase) to decompose the ECM structure would be an efficient method. However, there is no AIE-PS based on this approach that has been reported. (2) Since albumin has a long circulation half-life and continuous uptake in tumor tissues, it can help enhance the EPR effect. (3) Using a carrier such as PEG-encapsulating drugs to form nanoparticles. There are several AIE-PSs with an association with albumin that have been reported [155,156,157,158]. 

Recently, Tang et al. first proposed a mitochondria-targeting two-photon PSs (TPABP-Ir) with AIE properties based on an Ir(III) structure to generate Type I and Type II ROS (Figure 23) [156]. TPABP-Ir was coated with BSA to form TPABP-Ir@BSA nanoparticles (Ir-NPs). Colocalization imaging showed that the Pearson coefficient of Ir-NPs with mitochondria reached 0.86, indicating specific targeting of mitochondria. They also used DCFH as an indicator to detect total ROS production, and the fluorescence of the group treated with TPABP-Ir was 10 times higher than RB, and 17 times higher than Ce6. In addition, a significant increase in ROS production was observed in MCF-7 cells treated with Ir-NPs compared with the control cells, indicating a good ROS generation capacity. For in vivo experiments, the Annexin V-FITC and MTT assays demonstrated a good inhibitory ability of Ir-NPs by inducing apoptosis. Similar to in vitro experiments, Ir-NPs significantly inhibited tumor growth in tumor-bearing mice without obvious systemic toxicity. 

In 2021, Liu et al. [159] reported a new coordination polymer nanoparticle (CPN) which could achieve synchronous radiotherapy (RT) and radiodynamic therapy (RDT) under X-ray irradiation (Figure 24). They synthesized Hf-AIE-PEG-DBCO nanoparticles with a significant tumor inhibition effect by encapsulating TPEDC-DAC (AIE-PSs) using PEG modified with DBCO (dibenzocyclooctyne). Bioorthogonal click chemistry was performed by adding the metabolic precursor Ac_4_ManNAz (used to modify azide groups on cell membrane glycans) and DBCO-modified PEG to enhance the accumulation and prolong the retention of CPNs in tumors. Compared with the control group, Hf-AIE-PEG-DBCO exhibited a more obvious increase in fluorescence of DCFH-DA pretreated with Ac4ManNAz under X-ray irradiation, which is consistent with better tumor inhibition in vitro. Hf-AIE-PEG-DBCO showed a strong inhibitory effect on tumor growth in vivo as well as in vitro. H&E staining of tumor tissues showed significant tumor death after co-incubation with Ac_4_ManNAz and HF-AIE-PEG-DBCO under light. In this study, the combination of radiotherapy and RDT had a significant killing effect following intravenous injection of CPNs, due to the high penetration of X-rays and the DBCO-mediated bioorthogonal click chemistry. 

Liu et al. also developed another example of PEG-modified nanoparticles of variable size [160]. NPs with a size of 100–200 nm had a better tumor enrichment effect through the EPR effect, while smaller NPs (<50 nm) had minimal adhesion to the extravasation site of tumor blood vessels and extracellular matrix, which favored intratumoral penetration. They developed Dox-PEG-PS@MIL-100 NPs for the pH-response of photosensitization and the nanoparticle size-reducing process (Figure 25). H_2_O_2_ could break down the tumor intake of Dox-PEG-PS@MIL-100 NPs and release TPABTDCT (AIE-PSs) for the activatable PDT process. Meanwhile, Dox-PEG can self-assemble into ultra-small nanoparticles (DOX NPs) that can penetrate deep into tumors. After that, Dox was released into the nucleus to damage DNA under a low-pH environment. It was proved that TPABTDCT has better ^1^O_2_ generation efficiency than Ce6. This work achieved advanced photodynamic–chemotherapy combination therapy.

### 3.2. Active Targeting

In recent years, scientists have become aware that EPR effects vary over time during tumor development, and that they are highly heterogeneous [145,161,162,163,164]. To enhance specificity, active targeting is increasingly preferred. Active targeting is primarily performed by modifying bioligands on the surface of NPs that have the ability to target specific receptors on tumor cells [149,165]. Current active targeting strategies mainly target growth factor receptors overexpressed in cancers of different tissue origins, such as folate (FA) [166], transferrin (Tf) receptor [167], epidermal growth factor receptor (EGFR) [168,169] and so on [170,171,172]. Active-targeting strategies for PDT are well-summarized in other review papers [172].

As tumors require more biotin than normal tissues, linking biotin units can be used to target tumors through overexpressed biotin receptors. In a recent work, Chen et al. introduced nitrobenzoic acid (TTVBA) and biotin units to synthesize AIE-PSs (TTVBA) that avoided fluorescence quenching caused by PET [173]. Moreover, the fluorescence enhancement was the highest that had ever been reported during the aggregation process. The colocalization experiment of HeLa cells demonstrates that Biotin-TTVBA was accumulated in the cytoplasm with a bright red color. Besides, Biotin-TTVBA also had high photobleaching resistance after 350 s irradiation. Additionally, the IC_50_ values for Biotin-TTVBA were 2.5, 2.5 and 10 μM for HeLa, MCF-7, and L-O2 cells, which means that Biotin-TTVBA could selectively kill tumor cells with a high expression of the biotin receptor.

Another example is using EGFR to achieve the ability to target tumors. DCTBT is a newly developed photosensitizer with AIE characteristics, which enables NIR-II (1000 nm) fluorescence imaging, Type-I PDT and photothermal therapy (PTT), as reported by Tang et al. (Figure 26) [174]. Amphiphilic polymers modified with an EGFR-targeting peptide were doped to encapsulate DCTBT. The ROS species results showed that the DCTBT produced only •OH and O_2_^•−^ through the Type I pathway. Besides, the DCTBT NP has comparable ROS production efficiency with rose bengal. The photothermal efficiency experiment showed that DCTBT NPs had better photothermal performance (59.6%) than CTBT NPs. In vivo testing of DCTBT NPs on PANC-1 tumors revealed that they remained fluorescent 48 h after injection. It was noted that the Target-NPs (with EGFR-targeting peptide) had a better anti-cancer effect than the non-Target-NPs (without EGFR-targeting peptide). The combination of Type I PDT-PTT and DCTBT significantly inhibited the growth of PANC-1 tumors in vitro and in vivo. This approach shows great promise for overcoming tumors in hypoxic environments.

In recent years, a series of new active-targeting methods have emerged, including targeting tumor tissues under hypoxic conditions by the anaerobic nature of some bacteria [175,176], such as Escherichia coli, or by using cell membrane camouflage or liposomes to target tumors [177,178,179,180,181]. In addition, using red blood cell membranes to disguise NPs can deceive the immune system and reduce the immune response to NPs. The use of stimuli-responsive PDT therapy such as ROS-responsive, pH-responsive, hypoxia-responsive, redox-responsive, and so on, is also a major research topic [182,183,184,185]. Simply put, a tumor-targeting strategy can effectively reduce the toxic and side effects of photosensitizers and improve the killing ability of tumors, especially hypoxic tumors.

Under hypoxic conditions, Tang developed a novel approach for addressing the problem of drug resistance by combining bacteria with the Type I PDT system of the TBP-2 (Figure 27) [186]. A PDT intervention of this kind has never been reported before, which has shown to significantly impair orthotopic colon cancer growth and overcome pre-treatment toxicity. PDT-mediated cancer treatment can be delivered effectively to hypoxic tumors, because TBP-2 contains two cationic structures enabling E. coli to absorb it into the periplasmic space. The authors found that AE had brighter pictures and mainly concentrated near the cell membrane. The co-localization experiment involving AE and a hypoxic area proved that AE appeared in a hypoxic environment. Similarly, another work by Tang et al. using TBPP with similar structure and uropathogenic Escherichia coli (UPEC) to induce urinary tract infection (UTI) also had the function of targeting hypoxic tumors and had a good effect on the treatment of bladder cancer [187].

Using PLT-derived vesicles (PV) from mouse blood samples, Tang et al. synthesized a biomimetic nano-enzyme (PMD) by wrapping DCPy and MnO_2_ nanoparticles (Figure 28) [188]. In the co-localization experiment, CT26 cell imaging showed that PMD was more closely related to cells. Subsequently, the authors found that CT26 cells treated with PMD had stronger fluorescence and higher Mn content at the same concentration compared with erythrocyte-membrane-coated MD (RMD). Compared with control, the group containing PMD had better ROS production (nearly 15-fold). Hypoxia-inducible-factor (HIF-1α) staining treated with PMD found that there was almost no HIF-1α. MTT assay of CT26 cells showed that PMD had a good and similar IC_50_ value regardless of hypoxia or normaxia. This indicates that it has a good inhibitory effect on tumor under hypoxic condition. A superior tumor-targeting effect of PMD particles was demonstrated by intravenous injection into CT26 tumor-bearing mice of RMD or PMD, and PMD accumulation was significantly higher than that of RMD. This work proved that molecules with PLT derived vesicles have better PDT efficacy than nanoparticles disguised by red blood cells alone, which provides guidance for the future development of related work.

## 4. Conclusions and Perspectives

As a new promising anti-tumor method, PDT has attracted wide attention. This review introduces organelle-targeting and tumor-targeting PSs based on AIE strategies from the past three years. We introduced the chemical structures of various PSs and their functions as well as the mechanisms of PSs to treat tumor cells. Additionally, we classified their targeting mechanisms and principles. Although the emergence of AIE strategies has solved the problem of poor efficacy of PDT in vivo to a certain extent, PDT still faces many problems at present [36,189]. However, the main challenges of PDT are not only limited to light intensity in tissues, tumor hypoxia, and low accumulation efficiency of PSs in tumors [190,191,192]. Future efforts should be devoted to the following aspects: (1) Developing AIE-based PSs targeting the less-reported organelles, such as the GA or nucleus, is worthy of investigation; (2) developing novel nanoparticle carries and targeting conjugates to modify AIE-PSs and improve their water solubility, tumor targeting and delivery efficiency [193]; (3) developing AIEgen-based PSs with NIR I or NIR II absorption and Type I PDT abilities to overcome the limited penetration depth and the drug resistance of tumors under hypoxia; (4) the construction of AIE-PSs with both tumor-targeting and organelle-targeting abilities to optimize the therapeutic performance; (5) smart AIEgen-based PSs which show stimuli-responsive abilities and combined therapeutic effects such as PDT, PTT, immunotherapy and sonodynamic therapy [193] are also highly desirable; (6) some tumor cells are resistant to one certain cell death mode, and therefore developing AIE-PSs that can cause multiple cell death pathways are appealing for the effective inhibition of these tumor cells.

## Figures and Tables

**Figure 1 biosensors-12-01027-f001:**
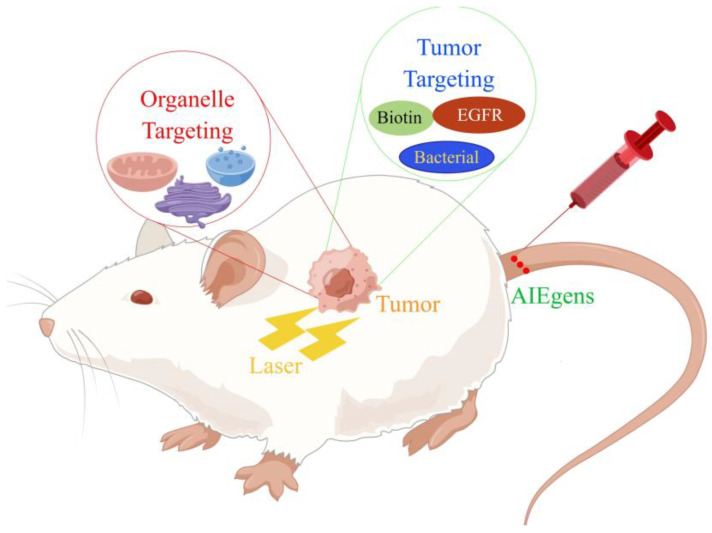
Different methods of treating tumor hypoxia.

**Figure 2 biosensors-12-01027-f002:**
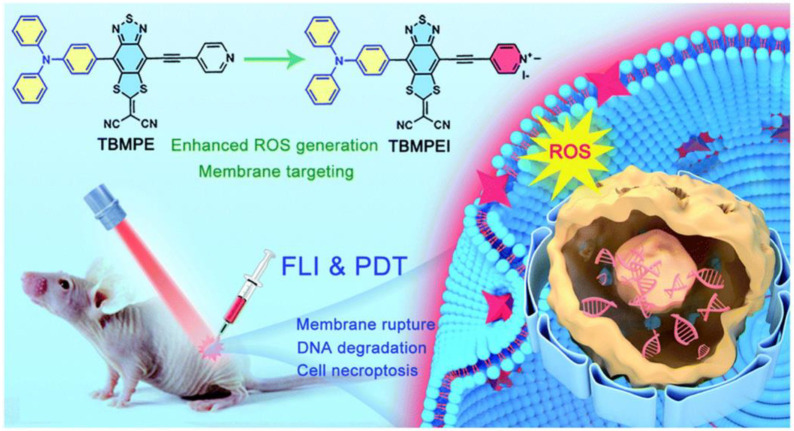
Schematic illustration of TBMPEI with cell-membrane-targeting function. Reprinted with permission from Ref. [41]. Copyright 2022 Royal Society of Chemistry.

**Figure 3 biosensors-12-01027-f003:**
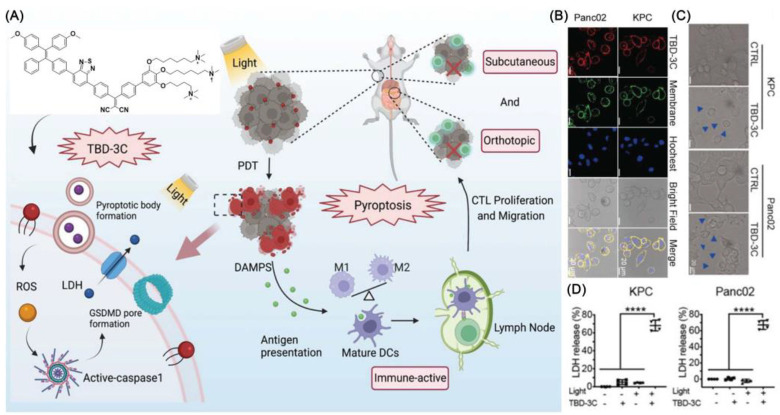
(**A**) Photodynamic pyroptosis for antitumor immunotherapy; (**B**) images of cells incubated with TBD-3C captured with CLSM; (**C**) KPC and Panc02 cells were analyzed by confocal microscopy after being exposed to 40 mW cm^−2^ for 10 min; bright blue arrows indicate membrane expansion; (**D**) Generated LDH from KPC and Panc02 cells irradiated with light at 40 W cm^−2^ for 10 min. (**** *p* < 0.0001).Reprinted with permission from Ref. [40]. Copyright 2022 WILEY-VCH.

**Figure 4 biosensors-12-01027-f004:**
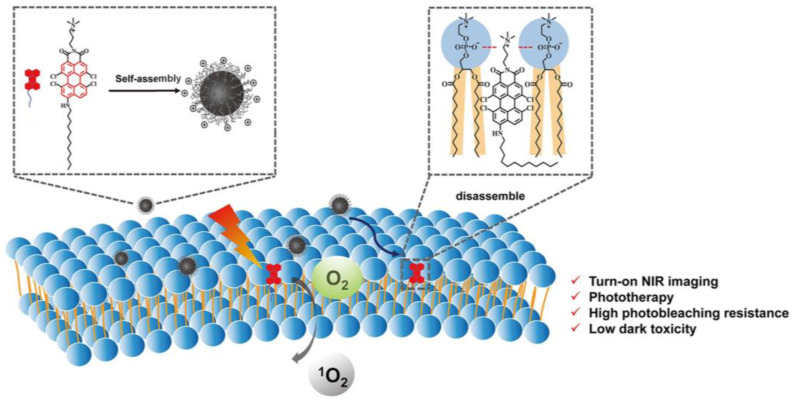
Schematic illustration of AP-based turn-on NIR fluorescence imaging and PDT. Reprinted with permission from Ref. [44]. Copyright 2021 American Chemical Society.

**Figure 5 biosensors-12-01027-f005:**
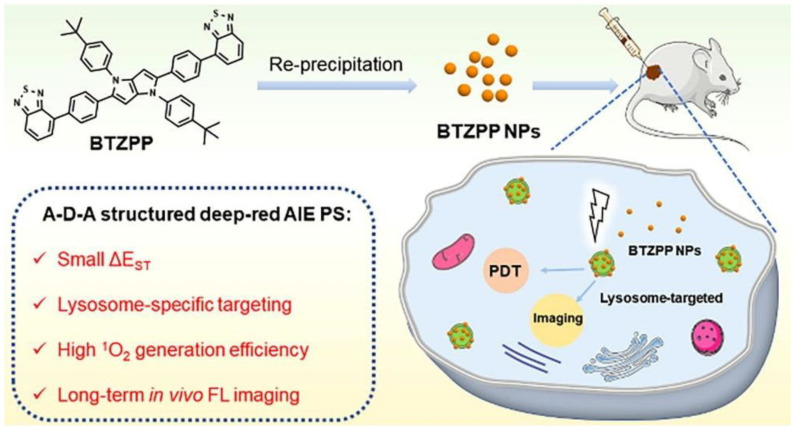
Schematic illustration of AIE-PSs (BTZPP) designed for long-term imaging. Reprinted with permission from Ref. [47]. Copyright 2022 Elsevier Ltd.

**Figure 6 biosensors-12-01027-f006:**
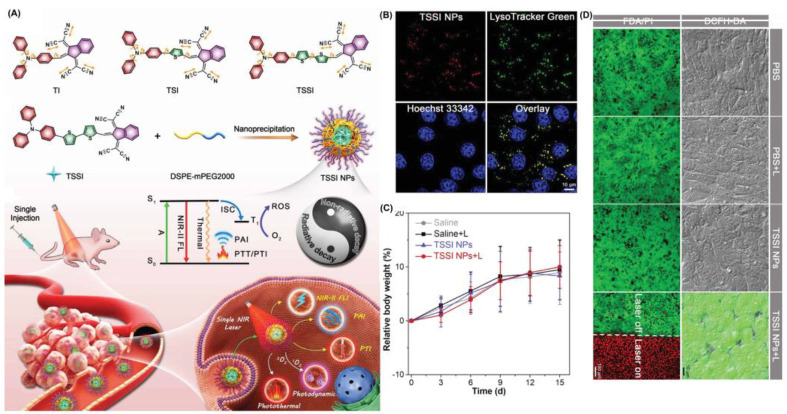
(**A**) Schematic illustration of mechanism and phototheranostic function of TSSI; (**B**) co-localization of 4T1 cells after incubation with TSSI NPs; (**C**) body weight changes of mice in vivo biosafety evaluation; (**D**) live/dead cell staining and intracellular ROS of 4T1 cells. Reprinted with permission from Ref. [70]. Copyright 2020 WILEY-VCH.

**Figure 7 biosensors-12-01027-f007:**
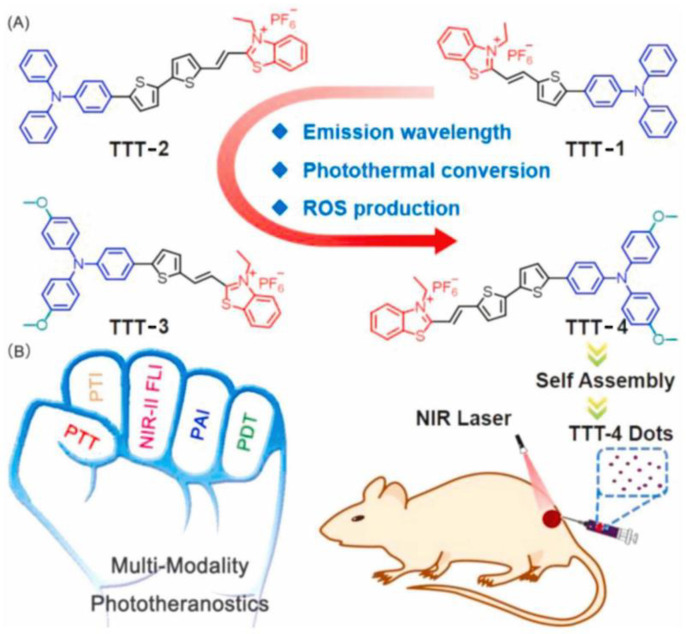
(**A**) Molecular structures and (**B**) multifunctional phototheranostic application of AIEgens. Reprinted with permission from Ref. [59]. Copyright 2021 Elsevier Ltd.

**Figure 8 biosensors-12-01027-f008:**
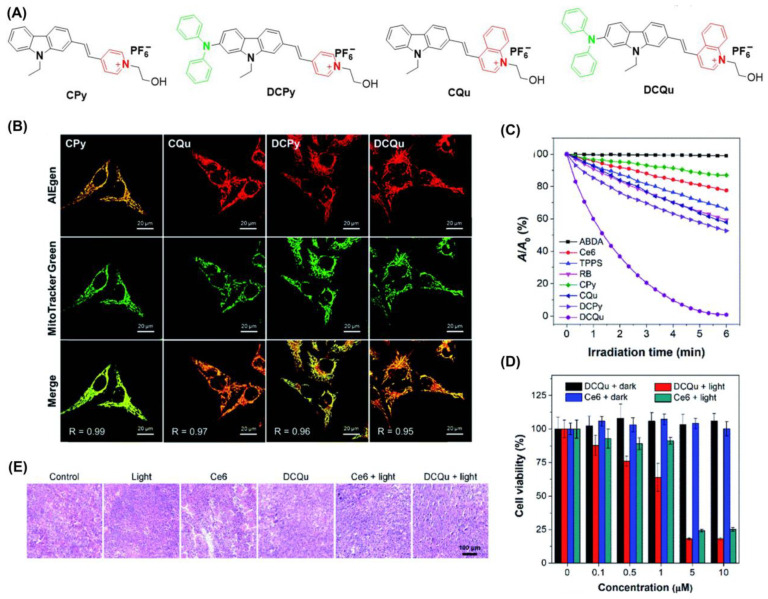
(**A**) Structure of CPy, CQu, DCPy and DCQu; (**B**) CLSM images of HeLa cells stained with CPy, CQu, DCPy and DCQu and various probes; (**C**) detection of ^1^O_2_ through ABDA; (**D**) cell viability of HeLa cancer cells stained with DCQu; (**E**) H&E staining analysis of tumor tissues treated with DCQu. Reprinted with permission from Ref. [75]. Copyright 2020 Royal Society of Chemistry.

**Figure 9 biosensors-12-01027-f009:**
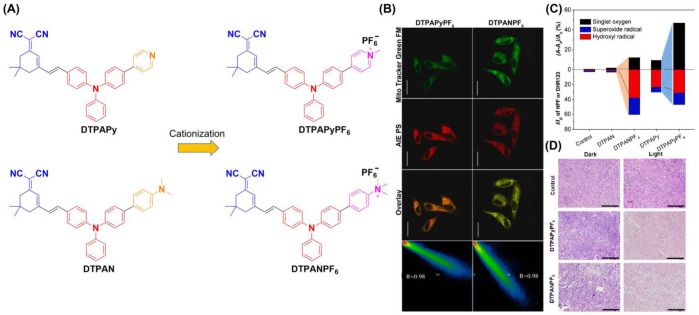
(**A**) Structure of DTPAN, DTPAPy, DTPANPF_6_ and DTPAPyPF_6_; (**B**) co-localization of HeLa cells incubated with DTPANPF_6_ or DTPAPyPF_6_; (**C**) summary of different ROS generation of DTPAN, DTPAPy, DTPANPF_6_ and DTPAPyPF_6_; (**D**) H&E staining of tumor tissues. Reprinted with permission from Ref [78]. Copyright 2022 Elsevier Ltd.

**Figure 10 biosensors-12-01027-f010:**
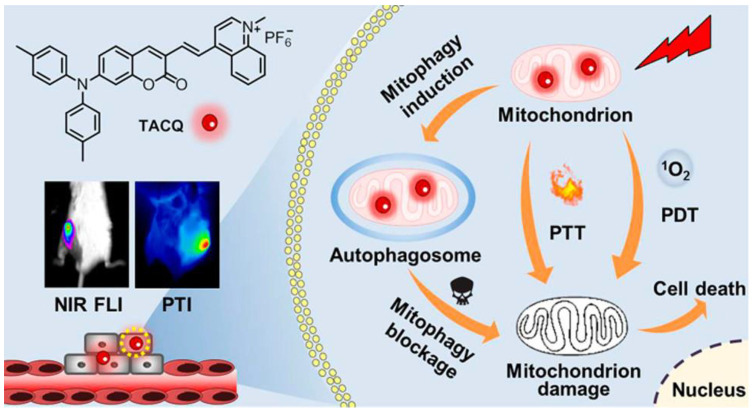
Schematic illustration of chemical structures and multifunctional phototheranostic of TACQ. Reprinted with permission from Ref. [90]. Copyright 2021 American Chemical Society.

**Figure 11 biosensors-12-01027-f011:**
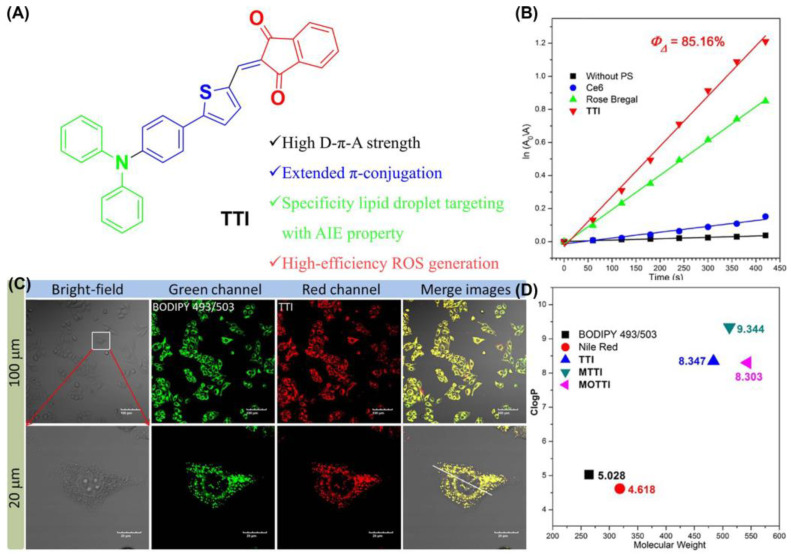
(**A**) Chemical structure and functions of TTI; (**B**) ^1^O_2_ quantum yield with TTI and Clog *p* values of TTI; (**C**) CLSM images of HepG2 cells stained with TTI and BODIPY 493/503; (**D**) Clog *p* values of TTI and its derivatives. Reprinted with permission from Ref. [119]. Copyright 2021 Elsevier Ltd.

**Figure 12 biosensors-12-01027-f012:**
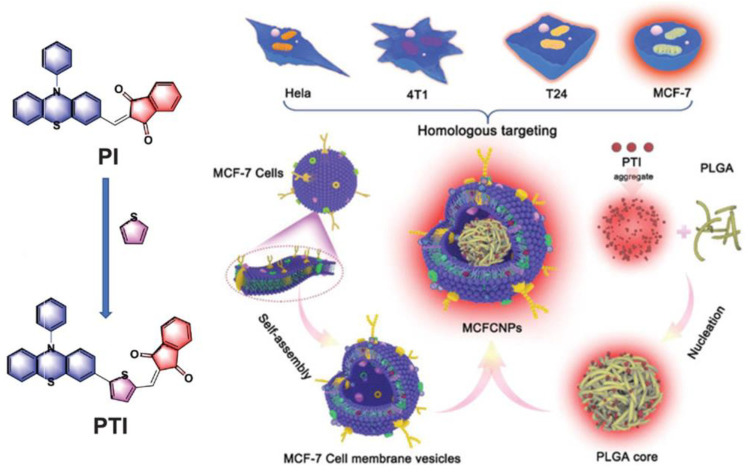
Schematic illustration of chemical structures and mechanisms of TACQ. Reprinted with permission from Ref. [124]. Copyright 2021 WILEY-VCH.

**Figure 13 biosensors-12-01027-f013:**
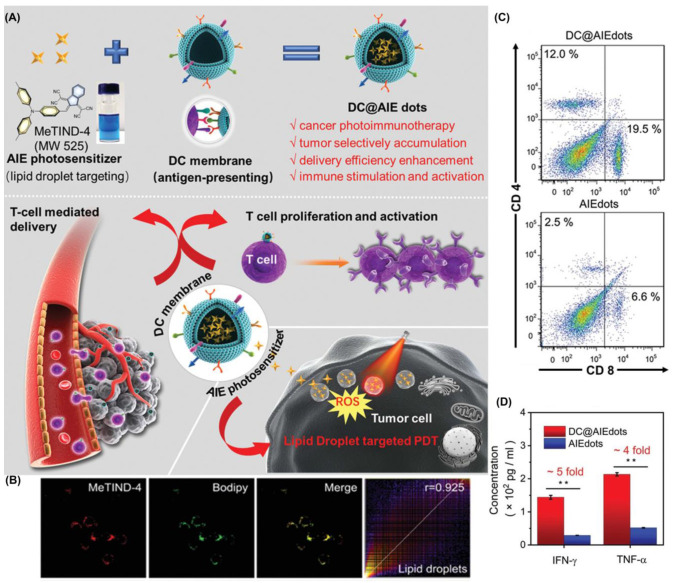
(**A**) Schematic illustration of chemical structures and in vivo photodynamic immunotherapy of DC@ AIEdots; (**B**) colocalization experiments of 4T1 cells stained with MeTIND-4; (**C**) flow cytometry analysis of tumor infiltrating CD8^+^ and CD4^+^ T cells; (**D**) concentration of proinflammatory cytokines TNF-α and IFN-γ. * *p* < 0.05, ** *p* < 0.01. Reprinted with permission from Ref. [114]. Copyright 2021 WILEY-VCH.

**Figure 14 biosensors-12-01027-f014:**
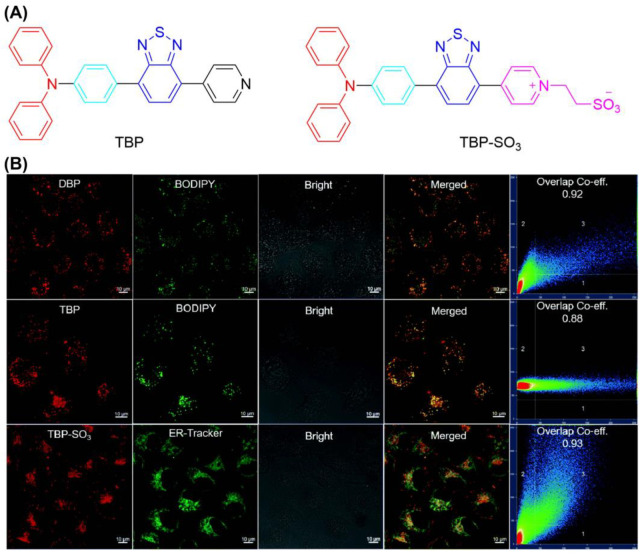
(**A**) Chemical structure of TBP and TBP-SO3; (**B**) CLSM images of HeLa cells stained with TBP-SO3. Reprinted with permission from Ref. [127]. Copyright 2022 WILEY-VCH.

**Figure 15 biosensors-12-01027-f015:**
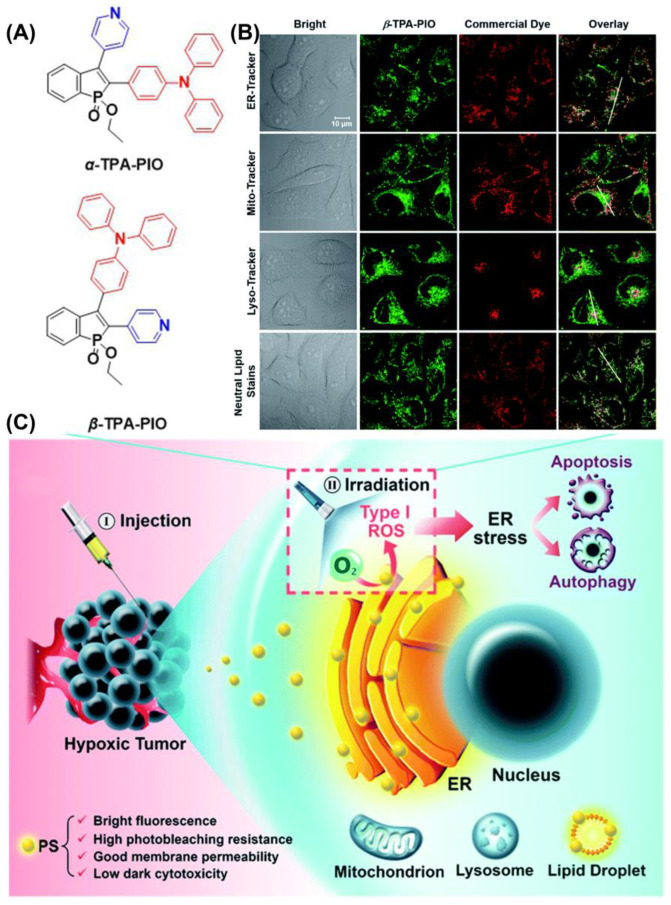
(**A**) Chemical structure of *α*-TPA-PIO and *β*-TPA-PIO; (**B**) CLSM images of HeLa cells co-stained with *β*-TPA-PIO; (**C**) schematic illustration of PDT treatment treated with PIO-based PSs. Reprinted with permission from Ref. [128]. Copyright 2020 Royal Society of Chemistry.

**Figure 16 biosensors-12-01027-f016:**
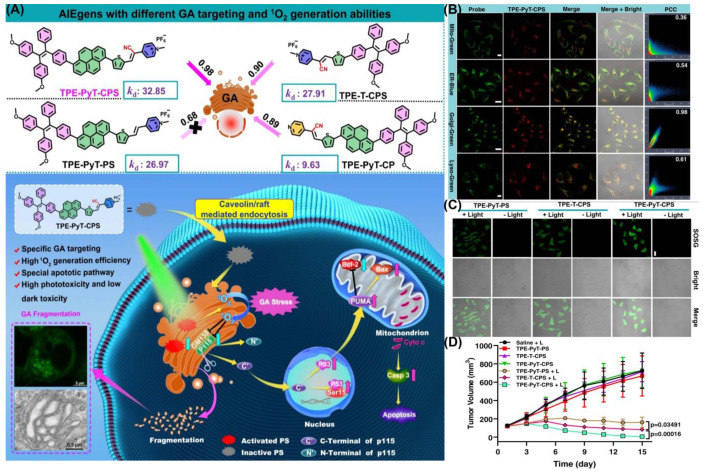
(**A**) Schematic illustration of TPE-PYT-CPS-induced GA stress and induced cell apoptosis upon PDT; (**B**) CLSM of HeLa cells stained with TPE-PYT-CPS and different probes; (**C**) intracellular ^1^O_2_ detection by CLSM; (**D**) changes of tumor volume treated with different AIEgens. Reprinted with permission from Ref. [129]. Copyright 2022 Nature Communication.

**Figure 17 biosensors-12-01027-f017:**
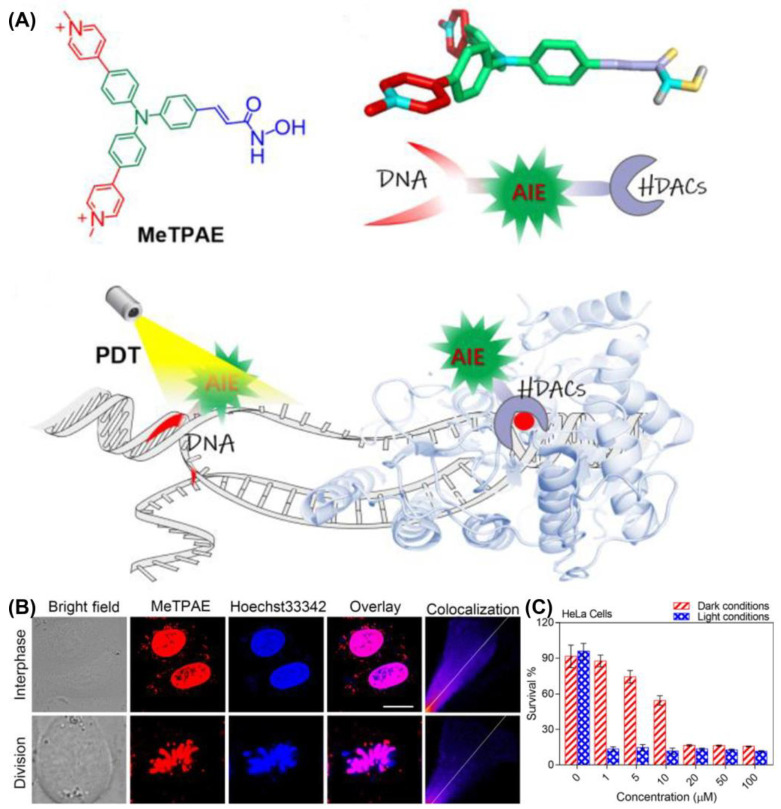
(**A**) Schematic illustration of chemical structure and mechanisms of MeTPAE with nucleic acids and histone deacetylase; (**B**) colocalization images of MeTPAE and Hoechst 33342 in HeLa cells during cell interphase or division; (**C**) cell viability of HeLa cell after being treated with MeTPAE. Reprinted with permission from Ref. [130]. Copyright 2022 WILEY-VCH.

**Figure 18 biosensors-12-01027-f018:**
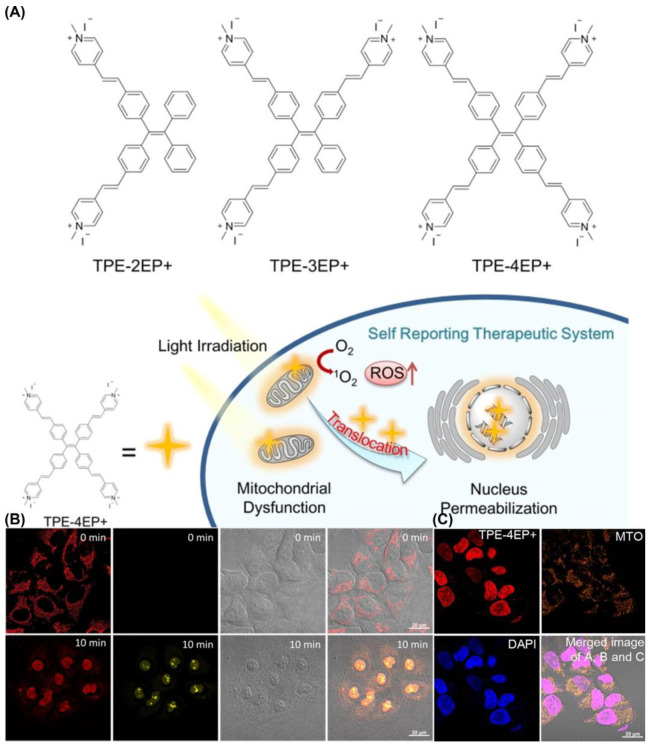
(**A**) Molecular structures and mechanism of mitochondria-to-nucleus translocation of TPE-4EP+; (**B**) mitochondria-to-nucleus translocation images of HeLa Cells stained with TPE-4EP+; (**C**) colocalization experiments of HeLa Cells co-incubation with TPE-4EP+. Reprinted with permission from Ref. [131]. Copyright 2019 American Chemical Society.

**Figure 19 biosensors-12-01027-f019:**
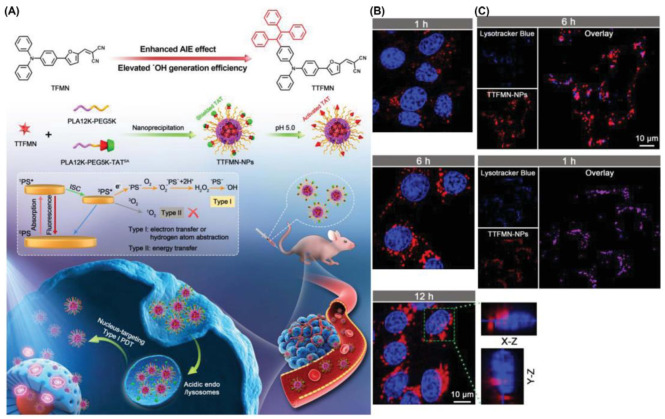
(**A**) Molecular structures and multifunctional phototheranostic application of TTFMN-NPs; (**B**) intracellular tracking on 4T1 cells; (**C**) CLSM images of nuclear targeting delivery on 4T1 cells. Reprinted with permission from Ref. [61]. Copyright 2021 WILEY-VCH.

**Figure 20 biosensors-12-01027-f020:**
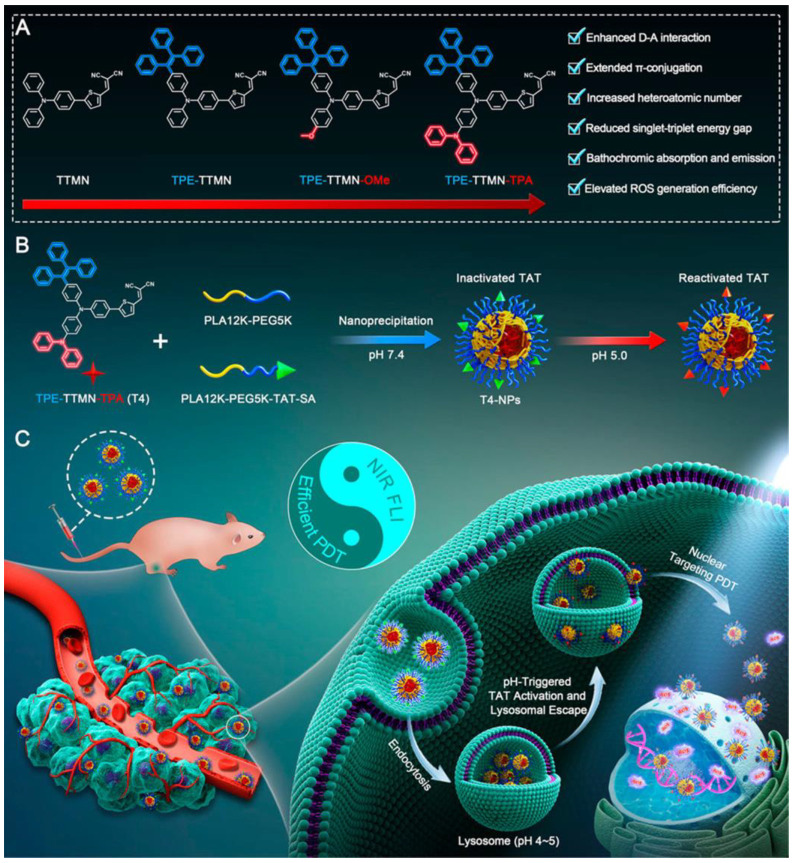
Schematic illustration of (**A**) molecular engineering of TPE-TTMN-TPA, (**B**) construction of nucleus-targeted T4-NPs, and (**C**) applications in PDT. Reprinted with permission from Ref. [60]. Copyright 2021 American Chemical Society.

**Figure 21 biosensors-12-01027-f021:**
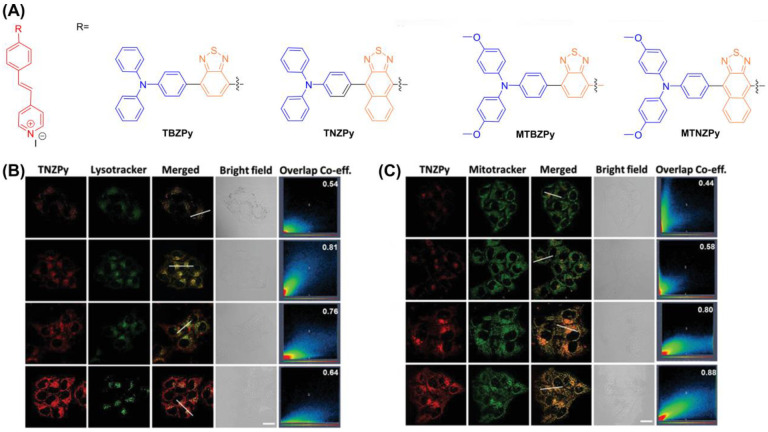
(**A**) Chemical structure of TBZPy, MTBZPy, TNZPy, MTNZPy; (**B**) CLSM images of HeLa cells stained with TNZPy and Lysotracker green; (**C**) CLSM images of HeLa cells stained with TNZPy and Mitotracker green. Reprinted with permission from Ref. [138]. Copyright 2020 WILEY-VCH.

**Figure 22 biosensors-12-01027-f022:**
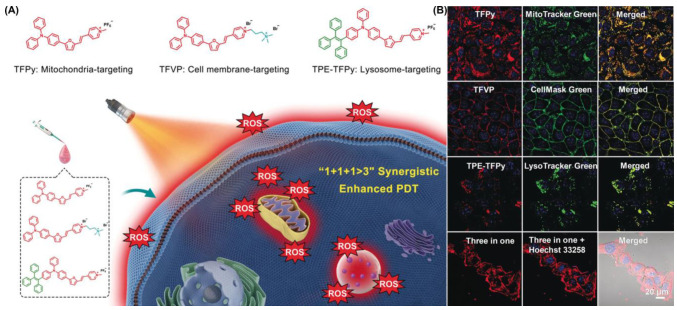
(**A**) Schematic illustration using TFPy, TFVP and TPE-TFPy for achieving 1 + 1 + 1 > 3 synergistic enhanced PDT; (**B**) co-localization of these three AIEgens. Reprinted with permission from Ref. [41]. Copyright 2020 WILEY-VCH.

**Figure 23 biosensors-12-01027-f023:**
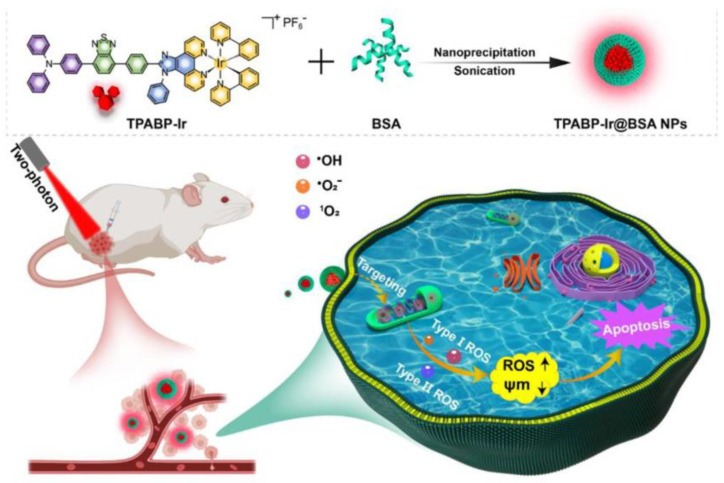
Schematic illustration of TPABP-Ir@BSA NPs for PDT treatment. Reprinted with permission from Ref. [156]. Copyright 2022 Elsevier Ltd.

**Figure 24 biosensors-12-01027-f024:**
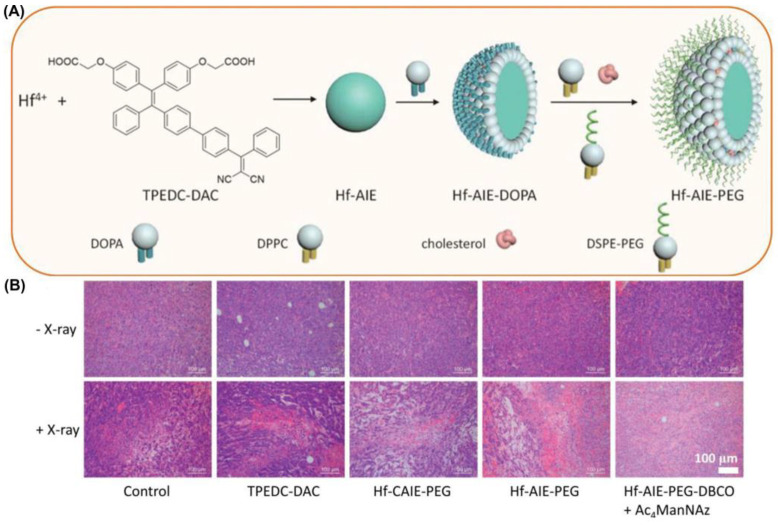
(**A**) Schematic illustration of fabrication of Hf-AIE-PEG; (**B**) H&E staining analysis of tumor tissues treated with various treatments. Reprinted with permission from Ref. [159]. Copyright 2021 WILEY-VCH.

**Figure 25 biosensors-12-01027-f025:**
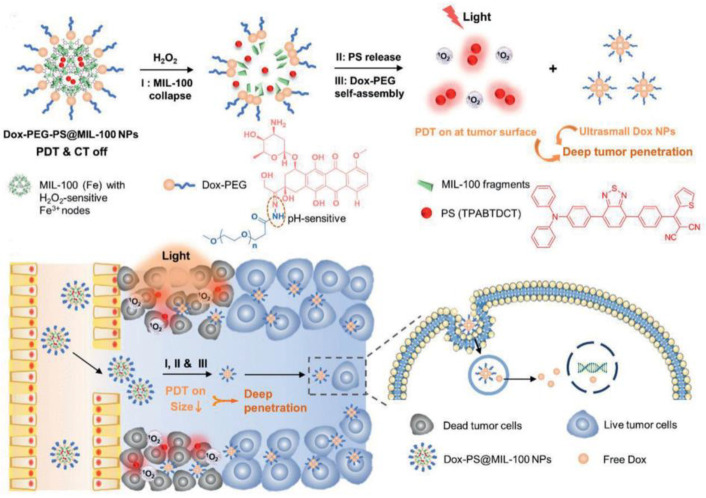
Schematic illustration of Dox-PEG-PS@MIL-100 NPs for advanced photodynamic–chemotherapy combination therapy. Reprinted with permission from Ref. [160]. Copyright 2021 WILEY-VCH.

**Figure 26 biosensors-12-01027-f026:**
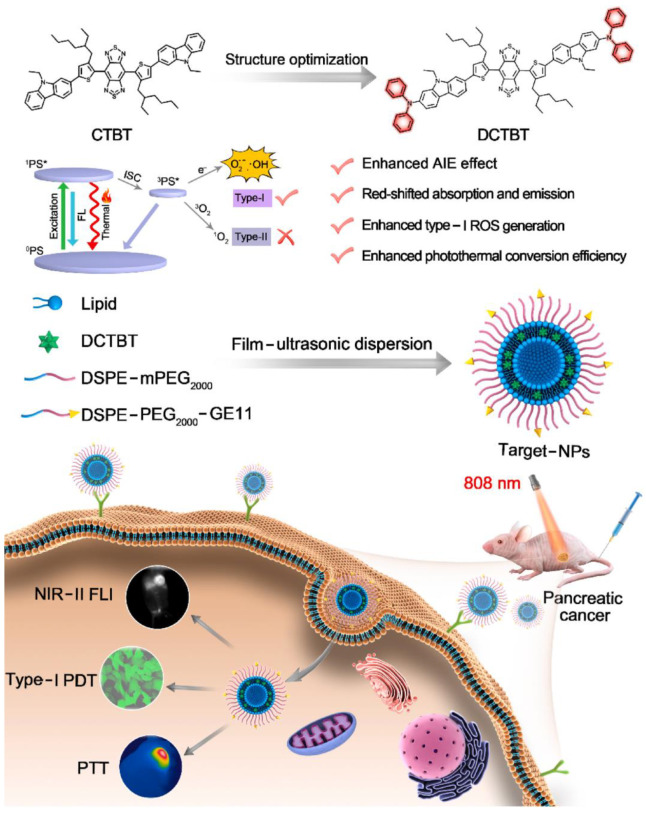
Schematic illustration of DCTBT, and the application on NIR-II FLI-guided Type-I PDT-PTT. Reprinted with permission from Ref. [174]. Copyright 2022 Elsevier Ltd.

**Figure 27 biosensors-12-01027-f027:**
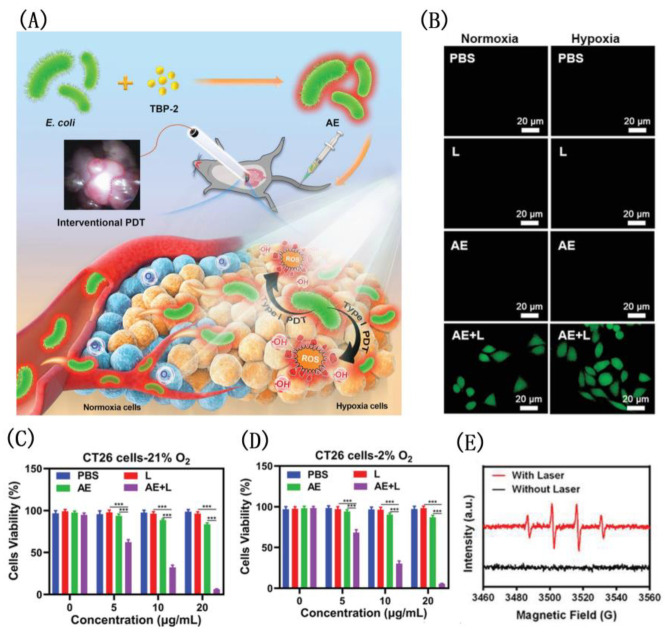
(**A**) The structure of TBP-2; viability of CT26 cells under normoxia (**B**) and (**C**) hypoxicconditions; (**D**) CLSM image of intracellular ROS; (**E**) schematic illustration of bright bacteria with TBP-2 for photodynamic therapy. Reprinted with permission from Ref. [186]. Copyright 2021 WILEY-VCH.

**Figure 28 biosensors-12-01027-f028:**
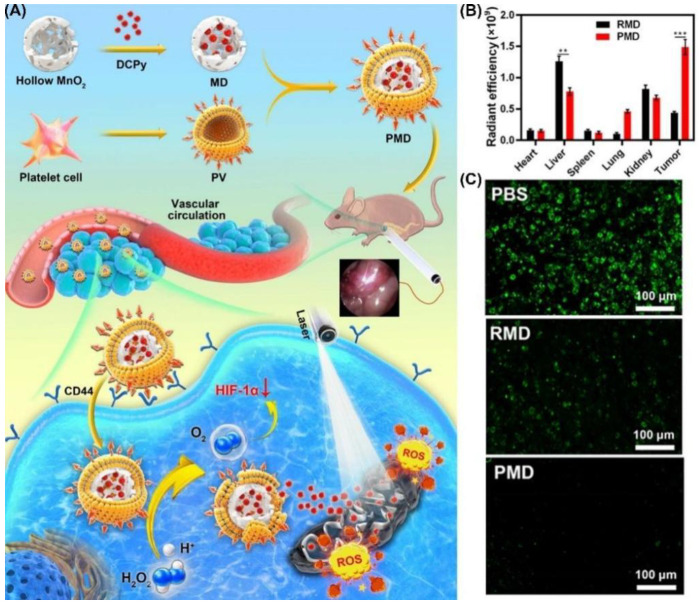
(**A**) Schematic illustration of the use of DCPy and PV as bionic nanozymes for photodynamic therapy; (**B**) radiant efficiency in tumors and different organs after PMD or RMD injection; (**C**) fluorescent staining for HIF-1α (representing intratumoral hypoxia). ** *p* < 0.01, *** *p* < 0.001. Reprinted with permission from Ref. [188]. Copyright 2022 American Chemical Society.

## Data Availability

Not applicable. No new data were created or analyzed in this study.

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
