# Peer review of "Aggregation-Induced Emission Luminogens for Enhanced Photodynamic Therapy: From Organelle Targeting to Tumor Targeting"

_biosensors, 2022, doi:10.3390/bios12111027_

Round 1
Reviewer 1 Report
This review mainly introduced photodynamic therapy (PDT) as a new promising anti-tumor method. Taking material as a starting point, aggregation-induced emission luminogen (AIEgens) based photosensitizers (PSs) showed enhanced reactive oxygen species (ROS) generation upon aggregation and showed great potential for hypoxic tumor treatment with enhanced PDT effect. AIEgens based PSs for enhanced PDT were summarized in two categories: organelle-targeting and tumor-targeting PSs. In each case, the authors tried their best to elaborate a definite molecular design for desired targeting. As a whole, this review will attract the readers’ attention to AIEgens based PSs for PDT. I recommend the publication of this review in Biosensors, but some concerns need to be addressed.
(1) The authors have concluded that PSs generated ROS through two processes of Type I and Type II. However, some sections mentioned which Type was for the generated ROS, while another sections vaguely introduced the ROS. This issue needs to be considered carefully for providing the readers with a clear guidance.
(2) Can the authors summarize a clear design to develop multiple organelle-targeting AIE-based PSs?
Author Response
Reviewer 1: This review mainly introduced photodynamic therapy (PDT) as a new promising anti-tumor method. Taking material as a starting point, aggregation-induced emission luminogen (AIEgens) based photosensitizers (PSs) showed enhanced reactive oxygen species (ROS) generation upon aggregation and showed great potential for hypoxic tumor treatment with enhanced PDT effect. AIEgens based PSs for enhanced PDT were summarized in two categories: organelle-targeting and tumor-targeting PSs. In each case, the authors tried their best to elaborate a definite molecular design for desired targeting. As a whole, this review will attract the readers’ attention to AIEgens based PSs for PDT. I recommend the publication of this review in Biosensors, but some concerns need to be addressed.
Response: Thanks for your comments and recognition. We have provided point-by-point response as follows.
- The authors have concluded that PSs generated ROS through two processes of Type I and Type II. However, some sections mentioned which Type was for the generated ROS, while another sections vaguely introduced the ROS. This issue needs to be considered carefully for providing the readers with a clear guidance.
Response: Thanks for your comments. We described PSs with Type I ROS generation capacity additionally in each example, while those that are not mentioned are Type II or not described by the author in the original text.
- Can the authors summarize a clear design to develop multiple organelle-targeting AIE-based PSs?
Response: Thanks for your careful reminding. At present, there are few reports on multi-organelle targeting AIEgens[1-9], so it is difficult to summarize a unified targeting strategy based on the reported literature. For the parts of using different PSs to achieve targeting various organelles[1-3], We propose that their different cellular uptake pathways and energy barriers to cell membranes account for their targeting differences. For another part of targeting different organelles achieved by using a single PSs[4-6, 8, 9], we hypothesized that a certain degree of lipophilicity and cation in the structure design would facilitate better targeting to mitochondria and lysosomes. Some of these reported articles are based on such logic to achieve multi-organelle targeting[4-6, 9].
(1) Feng, G. X.; Wang, C.; Chen, C. J.; Pan, Y. T.; Wu, M.; Wang, Y. B.; Liu, J.; Liu, B. Modulating Cell Specificity and Subcellular Localization by Molecular Charges and Lipophilicity. Chemistry of materials 2020, 32 (24), 10383-10393.
(2) Ni, J. S.; Lee, M. M. S.; Zhang, P.; Gui, C.; Chen, Y.; Wang, D.; Yu, Z. Q.; Kwok, R. T. K.; Lam, J. W. Y.; Tang, B. Z. SwissKnife-Inspired Multifunctional Fluorescence Probes for Cellular Organelle Targeting Based on Simple AIEgens. Anal Chem 2019, 91 (3), 2169-2176.
(3) Chen, K. Q.; Zhang, R. Y.; Wang, Z. M.; Zhang, W. J.; Tang, B. Z. Structural Modification Orientated Multifunctional AIE Fluorescence Probes: Organelles Imaging and Effective Photosensitizer for Photodynamic Therapy. Advanced Optical Materials 2020, 8 (2).
(4) Wan, Q.; Zhang, R.; Zhuang, Z.; Li, Y.; Huang, Y.; Wang, Z.; Zhang, W.; Hou, J.; Tang, B. Z. Molecular Engineering to Boost AIE‐Active Free Radical Photogenerators and Enable High‐Performance Photodynamic Therapy under Hypoxia. Advanced functional materials 2020, 30 (39).
(5) Chai, C.; Zhou, T.; Zhu, J.; Tang, Y.; Xiong, J.; Min, X.; Qin, Q.; Li, M.; Zhao, N.; Wan, C. Multiple Light-Activated Photodynamic Therapy of Tetraphenylethylene Derivative with AIE Characteristics for Hepatocellular Carcinoma via Dual-Organelles Targeting. Pharmaceutics 2022, 14 (2).
(6) Wang, K. N.; Liu, L. Y.; Mao, D.; Hou, M. X.; Tan, C. P.; Mao, Z. W.; Liu, B. A Nuclear-Targeted AIE Photosensitizer for Enzyme Inhibition and Photosensitization in Cancer Cell Ablation. Angew Chem Int Ed Engl 2022, 61 (15), e202114600.
(7) Kang, M.; Zhang, Z.; Xu, W.; Wen, H.; Zhu, W.; Wu, Q.; Wu, H.; Gong, J.; Wang, Z.; Wang, D.; et al. Good Steel Used in the Blade: Well-Tailored Type-I Photosensitizers with Aggregation-Induced Emission Characteristics for Precise Nuclear Targeting Photodynamic Therapy. Advanced Science 2021, 8 (14).
(8) Zhang, Z.; Xu, W.; Xiao, P.; Kang, M.; Yan, D.; Wen, H.; Song, N.; Wang, D.; Tang, B. Z. Molecular Engineering of High-Performance Aggregation-Induced Emission Photosensitizers to Boost Cancer Theranostics Mediated by Acid-Triggered Nucleus-Targeted Nanovectors. ACS Nano 2021, 15 (6), 10689-10699.
(9) Zhang, T.; Li, Y.; Zheng, Z.; Ye, R.; Zhang, Y.; Kwok, R. T. K.; Lam, J. W. Y.; Tang, B. Z. In Situ Monitoring Apoptosis Process by a Self-Reporting Photosensitizer. J. Am. Chem. Soc. 2019, 141 (14), 5612-5616.
Reviewer 2 Report
In this review, Guo and coworkers have highlighted the application of Aggregation-induced emission luminogen (AIEgens) based photosensitizers (PSs) for photodynamic therapy. Recent research advancements with respect to organelle-targeting AIEgens and tumor-targeting AIEgens have been briefly summarized. The topic is of great interest in both biomedical engineering and materials science. The review described the state-of-the-art area with clear logic and concise language. Hence, I recommend the publication of this review in Biosensors after the authors address the following minor issues:
1. For better comparison, the authors may provide the generation efficiency of reactive oxygen species upon irradiating the AIEgens with light. In some cases, the detail was missing.
2. Some words in Figures 3, 9, 11, 13, 14, 16, 19, 20, 24, and 28 are unreadable, and the structural formulas of AIEgens in Figures 3, 13, and 20 are a bit vague. It would be better to make them clearer. Please double-check the readability of all Figures.
3. The author had better summarize several factors in designing AIE-based photosensitizers to get a better photodynamic therapy effect. And the author should conclude which structural feature AIEgen should have in order to get active/passive targeting.
Author Response
Reviewer 2: In this review, Guo and coworkers have highlighted the application of Aggregation-induced emission luminogen (AIEgens) based photosensitizers (PSs) for photodynamic therapy. Recent research advancements with respect to organelle-targeting AIEgens and tumor-targeting AIEgens have been briefly summarized. The topic is of great interest in both biomedical engineering and materials science. The review described the state-of-the-art area with clear logic and concise language. Hence, I recommend the publication of this review in Biosensors after the authors address the following minor issues:
1.For better comparison, the authors may provide the generation efficiency of reactive oxygen species upon irradiating the AIEgens with light. In some cases, the detail was missing.
Response: Thanks for your careful reminding. We supplemented each article with additional references to ROS production efficiency. Some articles did not elaborate on the singlet oxygen production efficiency, so we used other methods to supplement it, such as the degradation rate of ABDA and the fluorescence enhancement rate of DCFH, etc.
2.Some words in Figures 3, 9, 11, 13, 14, 16, 19, 20, 24, and 28 are unreadable, and the structural formulas of AIEgens in Figures 3, 13, and 20 are a bit vague. It would be better to make them clearer. Please double-check the readability of all Figures.
Response: Thanks for your careful reminding. We have revised the above unclear or misleading pictures.
3.The author had better summarize several factors in designing AIE-based photosensitizers to get a better photodynamic therapy effect. And the author should conclude which structural feature AIEgen should have in order to get active/passive targeting.
Response: Thanks for your careful reminding. Passive targeting, as mentioned in the article, is mainly based on the EPR effect produced by nanoparticles. The enhancement of EPR effect is mainly through the modification of nanoparticles with substances that can have better permeability in the tumor microenvironment. However, active targeting is achieved by binding to PS groups or carriers that themselves have active targeting ability. The structures and targeting strategies of these groups and vectors are well studied. These strategies have also been mentioned in the article
Reviewer 3 Report
This review reports on the latest results in aggregation-induced emission luminogens for PDT. The manuscript is well organized in the different section depending of the organelle-targeting AIE-PDT. However, in spite that the authors have been done a good work compiling all recent examples, they only summarize one after the other the previous examples. The graphics come from the original papers and in many cases, they are not of enough quality or the are hardly seem. They have put together several images from the original papers and them is difficult to follow.
Consequently, although the review may be of interest for the readers, I recommend unless to modify the graphics or separate some of the drawing in the search of clarity.
Author Response
Reviewer 3: This review reports on the latest results in aggregation-induced emission luminogens for PDT. The manuscript is well organized in the different section depending of the organelle-targeting AIE-PDT. However, in spite that the authors have been done a good work compiling all recent examples, they only summarize one after the other the previous examples. The graphics come from the original papers and in many cases, they are not of enough quality or the are hardly seem. They have put together several images from the original papers and them is difficult to follow.
1.Consequently, although the review may be of interest for the readers, I recommend unless to modify the graphics or separate some of the drawing in the search of clarity.
Response: Thanks for your comments and recognition. We have changed the picture and re-described it. Please review it again.
A list of changes:
- For the the Figures in the revised manuscript, we have revised all the scale bars included were reprocessed to ensure them visible.
- We added explanation into several parts of the article
We added the briefly mention of an additional elaboration in row 45-48
We classified in the first segment of multi-organelle targeting in row 383-385
- We changed the order of the third (page 17) and fourth (page 18) examples and their pictures in the multi-organelle targeting section to make this section easier to understand.
In most of our examples, we have added a specific description of the efficiency of ROS production on page 3-21,23
Round 2
Reviewer 3 Report
The authors have made all the changes required and now it can be accepted.